# Luminescence Properties of an Orthorhombic KLaF_4_ Phosphor Doped with Pr^3+^ Ions under Vacuum Ultraviolet and Visible Excitation

**DOI:** 10.3390/ma17061410

**Published:** 2024-03-19

**Authors:** Patrycja Zdeb, Nadiia Rebrova, Radosław Lisiecki, Przemysław Jacek Dereń

**Affiliations:** Institute of Low Temperature and Structure Research, Polish Academy of Science, Okólna Street 2, 50-422 Wrocław, Poland; p.zdeb@intibs.pl (P.Z.); n.rebrova@intibs.pl (N.R.); r.lisiecki@intibs.pl (R.L.)

**Keywords:** KLaF_4_, VUV spectroscopy, quantum cutting, fluoride phosphor, luminescence thermometry

## Abstract

Fluorides have a wide bandgap and therefore, when doped with the appropriate ions, exhibit emissions in the ultraviolet C (UVC) region. Some of them can emit two photons in the visible region for one excitation photon, having a quantum efficiency greater than 100%. In a novel exploration, praseodymium (Pr^3+^) ions were introduced into KLaF_4_ crystals for the first time. The samples were obtained according to a high-temperature solid-state reaction. They exhibited an orthorhombic crystal structure, which has not been observed for this lattice yet. The optical properties of the material were investigated in the ultraviolet (UV) and visible ranges. The spectroscopic results were used to analyze the Pr^3+^ electronic-level structure, including the 4*f*5*d* configuration. It has been found that KLaF_4_:Pr^3+^ crystals exhibit intense luminescence in the UVC range, corresponding to multiple 4*f* → 4*f* transitions. Additionally, under vacuum ultraviolet (VUV) excitation, distinct transitions, specifically ^1^S_0_ → ^1^I_6_ and ^3^P_0_ → ^3^H_4_, were observed, which signifies the occurrence of photon cascade emission (PCE). The thermal behavior of the luminescence and the thermometric performance of the material were also analyzed. This study not only sheds light on the optical behavior of Pr^3+^ ions within a KLaF_4_ lattice but also highlights its potential for efficient photon management and quantum-based technologies.

## 1. Introduction

In recent years, the exploration of rare-earth-doped crystals has significantly advanced the realm of photonics and optical technologies. Praseodymium (Pr), a member of the lanthanide series, intrigues researchers with its distinctive spectroscopic properties despite possessing only two 4*f* electrons. Notably, phosphors activated by Pr^3+^ ions have undergone extensive investigation across diverse fields, such as light sources [1,2,3], optical thermometry [4,5,6], laser technologies [7,8], glass filters [9,10] and bioimaging [11]. The optical properties of these kinds of materials in the UV range are particularly interesting because they can be modulated by changing the crystallographic surroundings of the activator ion. This is possible due to their 4*f*5*d* levels, which have an energy of about 61,580 cm^−1^ [12] in free Pr^3+^ ions. However, upon doping into crystals, these levels are split and shifted towards lower energies. This red shift significantly depends on the crystal structure type and the material composition hosting the Pr^3+^ ions. Consequently, under excitation using UV light, three distinct scenarios emerge [13]:If *E*(4*f*5*d*) < *E*(^1^S_0_), only interconfigurational 4*f*5*d* → 4*f* transitions are observed;If *E*(4*f*5*d*) >> *E*(^1^S_0_), nonradiative relaxation between these two levels occurs, and only 4*f* → 4*f* transitions are observed;If *E*(4*f*5*d*) ≥ *E*(^1^S_0_), both 4*f*5*d* → 4*f* and 4*f* → 4*f* transitions take place.

The influence of the crystal field on Pr^3+^ 4*f*5*d* level positions has been studied by several researchers, including Dorenbos [12,14,15,16,17] and Srivastava [13].

The process observed in the second case is known as ‘photon cascade emission’ (PCE) or ‘quantum cutting’ (QC). In this phenomenon, the absorption of one high-energy photon is followed by a two-step emission process that generates two photons. As a consequence, a quantum efficiency higher than 100% is possible to obtain. PCE was initially observed independently by Sommerdijk et al. [18] and Piper et al. [19] in 1974, marking a groundbreaking discovery. Since then, phosphors exhibiting PCE have gained a lot of attention due to their possible application in lamps based on Xe discharge, competitive with Hg-discharge-based lamps [20]. In Pr^3+^-doped materials, PCE is observed as two characteristic transitions, ^1^S_0_ → ^1^I_6_ (around 400 nm) and ^3^P_0_ → ^3^H_4_ (around 480 nm), which can be detected after excitation to the 4*f*5*d* level. Because *E*(4*f*5*d*) > *E*(^1^S_0_) is the crucial condition for PCE occurrence, this process is highly dependent on the type of crystal lattice. Thus, photon cascade emission was reported for limited types of hosts, including fluorides and oxides such as YF_3_ [21], KMgF_3_ [22], LiSrAlF_6_ [23], SrAl_12_O_19_ [24], and LaMgB_5_O_10_ [25].

Crystals represented by the formula ALnF_4_ (A = Li^+^, Na^+^, K^+^, Ln = Y^3+^, La^3+^, Lu^3+^) have been studied as a promising host for RE^3+^ ions due to their low phonon energies, high optical transparency in the UV-visible range, and good chemical and photochemical stability. Among these fluorides, some doped with Pr^3+^ ions have exhibited noteworthy characteristics. For instance, LiLuF_4_:Pr^3+^ and LiYF4:Pr^3+^ show intense UVC luminescence, attributed to interconfigurational 4*f*5*d* → 4*f* transitions, while others like NaYF_4_:Pr^3+^ and KYF_4_:Pr^3+^ demonstrate luminescence due to the PCE process [26]. It is worth noting that KLaF_4_ crystals stand out due to their exceptionally low phonon energy (262 cm^−1^) [27], positioning them as ideal hosts for highly efficient upconverters and downconverters [28]. Because of their great mono dispersion, long luminescence lifetime, and high up- or downconversion efficiency, they may have versatile and promising applications as luminescent nano-biolabels. Recent work by Deo et al. reported KLaF_4_ nanoparticles co-doped with Eu^3+^, Er^3+^, and Yb^3+^ ions, allowing simultaneous excitation in the visible and NIR regions, resulting in upconversion and downconversion emissions concurrently [29]. This dual-mode approach presents valuable applications in bioimaging or information encryption. Additionally, Nd^3+^-doped KLaF_4_ nanoparticle colloidal solutions were proposed as NIR high-power liquid laser materials and amplifiers [30]. Despite extensive studies on KLaF_4_, the incorporation of Pr^3+^ ions into this fluoride matrix remains unexplored. Consequently, the optical properties of KLaF_4_:Pr^3+^ are still unknown.

In this work, KLaF_4_ powders doped with Pr^3+^ ions were synthesized for the first time. To prepare this material, a high-temperature solid-state reaction was applied, leading to an orthorhombic structure. To the best of our knowledge, KLaF_4_ crystals with this type of space group have not been reported yet since this fluoride usually crystallizes in the cubic or hexagonal phase [27]. Furthermore, we studied the optical properties of KLaF_4_ doped with different concentrations of Pr^3+^ ions at room and low temperatures, as well as the temperature dependence of luminescence and thermometric performance. A special focus was put on spectroscopic measurements in the deep UV range, which revealed the great potential of this system as a photon cascade emitter.

## 2. Materials and Methods

Nanoparticles KLaF_4_:x% mol Pr^3+^, x = 0.1, 0.5, 1, 1.5, 2 (or KLa_1−x_Pr_x_F_4_, x = 0.001, 0.005, 0.01, 0.015, 0.02) were prepared via high-temperature solid-state reactions. The raw materials LaF_3_ (99.99%), KF (99.9%), and PrF_3_ (99.99%) were mixed and carefully ground in an agate mortar for 15 min. The powders were sintered at 680 °C for 8 h in a reducing atmosphere at a gas flow rate of 10 l/h (N_2_ = 95%, H_2_ = 5%).

To obtain X-ray powder diffraction (XRPD) patterns, a Panalytical X’Pert PRO powder diffractometer with a copper K_α_ radiation source (*λ* = 1.54056 Å) was used. The morphology, composition, and mapping of the samples were investigated using an FE-SEM FEI Nova NanoSEM 230 (FEI Company, a part of Thermo Fisher Scientific, Waltham, MA, USA) equipped with an energy-dispersive X-ray spectrometer, the EDAX Genesis XM4. The SEM images were recorded at 5.0 kV in beam deceleration mode, which improves imaging parameters such as the resolution and contrast. In the case of the SEM-EDX measurements, a large area (250 μm × 200 μm) of the samples was scanned at 20 kV. The powder samples were included in the carbon resin (PolyFast Struers, Ballerup, Denmark) and then pressed using an automatic mounting press CitoPress-1 (Struers, Ballerup, Denmark) in order to obtain a large and flat area. Signals from three randomly selected areas were collected to ensure satisfactory statistical averaging. The particle size distribution histogram was calculated using ImageJ v1.53k software by collecting the size of 109 particles.

To examine the excitation and emission spectra in the UVC range, a VUV McPherson spectrometer equipped with a water-cooled deuterium lamp and a Hamamatsu photomultiplier R955P was utilized. An FLS1000 fluorescence spectrometer from Edinburgh Instruments, equipped with a xenon lamp, was employed for the excitation and emission spectra, as well as the decay profiles in the visible range. The same spectrometer connected to the Linkam THMS 600 Heating/Freezing Stage was used to perform temperature-dependent measurements.

## 3. Results and Discussion

### 3.1. Structure

The structure of the KLaF_4_ crystals has been studied and described in numerous papers [28,31,32,33]; however, only cubic and hexagonal types of lattices have been reported. In this work, new orthorhombically ordered microcrystals were synthesized via a high-temperature solid-state reaction. Figure 1a shows the XRPD spectra of pure and Pr^3+^-doped KLaF_4_. The XRPD patterns were indexed similarly to the orthorhombic structure of KCeF_4_ (SG: *Pnma*; ICSD file No. 23229 [34]), and no peaks corresponding to any other phase were observed. In this lattice, La^3+^ ions are coordinated by nine fluorine atoms, forming a so-called tricapped triangular prism (Figure 1b). The unit cell parameters of KLaF_4_:Pr^3+^ are provided in Appendix A. Notably, the lattice parameters and unit cell volume decrease with an increase in the activator concentration due to the smaller radius of Pr^3+^ ions (1.179 Å for CN = 9) compared to that of La^3+^ ions (1.216 Å for CN = 9) [35].

Figure 2a,b present SEM images of the KLaF_4_ grains at two different scales. The synthesized particles have an agglomerated, non-uniform shape with sharp edges. Based on the size distribution histogram (Figure 2c), the mean size of the grains was estimated to be 6.9 μm. The observed particle size and morphology are characteristic of the solid-state method of synthesis, and similar results were also obtained for LuPO_4_ powders prepared using the same synthesis method [36]. Figure 2d shows the energy-dispersive spectroscopy (EDS) spectrum for a KLaF_4_:1.5%Pr^3+^ sample. Emission peaks were observed at 0.8 and 4.7 keV for lanthanum, 3.3 keV for potassium, 0.7 keV for fluorine, and 5.1 eV for praseodymium. The inset in Figure 2d displays the weight and atomic percentages of the elements. The atomic percentage corresponds to the ratio of the stoichiometric value of the element in the formula to the sum of the stoichiometric values of all the elements, which confirms that the KLaF_4_ crystals were obtained with a new crystallographic structure. Furthermore, EDS mapping analysis confirmed the homogeneous distribution of the K, La, F, and Pr elements (refer to Appendix A).

### 3.2. Optical Properties

#### 3.2.1. VUV Excitation

Figure 3a presents the excitation spectrum of the KLaF_4_:1%Pr^3+^ powder monitored at 272 nm. To describe all the observed transitions, the experimental data were deconvoluted into five Gaussian-like components (blue solid lines in Figure 3), which can correspond to the transition from the ^3^H_4_ ground state of Pr^3+^ ions to higher-energy-lying 5*d* levels. The energies and Full Width at Half Maximum (FWHM) values of the detected bands are detailed in Appendix A. The broad band, commencing around 71,000 cm^−1^, could be associated with host absorption, though other potential origins cannot be excluded. Further detailed research is necessary to confirm the assignment of this band.

The 5*d*-level positions of the lanthanide ions are influenced by factors such as centroid shift (*ε_c_*), crystal field splitting (*ε_cfs_*), and redshift (*D*(A)) [14]. Dorenbos investigated these parameters across various crystalline lattices, including fluorides [14], chlorides [15], oxides [16], and aluminates [17].

The centroid shift (*ε_c_*) represents the energy difference between the average positions of the 5*d* levels in a free RE^3+^ ion and within a crystalline host. The *ε_c_* value is influenced by the coordinating ligand and is the smallest for fluoride matrices according to the nephelauxetic series:F^−^ < Cl^−^ < Br^−^ < I^−^ < O^2−^ < S^2−^.

Crystal field splitting (*ε_cfs_*) is the energy difference between the lowest and highest 5*d* components. It tends to increase with a decreasing coordination number. In the orthorhombic KLaF_4_ lattice, La^3+^ ions are coordinated by nine fluorine anions, resulting in small *ε_cfs_ (*11,592 cm^−1^). Both *ε_c_* and *ε_cfs_* contribute to the redshift (*D*(A)) of the first allowed 4*f* → 5*d* transition in host A. This redshift can be expressed as:(1)$$D\left(A\right)=E_{5d}\left(free\right)-E_{5d}\left(A\right),$$
where *E*_5*d*_(free) is the position of the lowest 5*d* level of RE^3+^ as a free ion (for Pr^3+^, *E*_5*d*_(free) = 61,580 cm^−1^) and *E_5d_*(A) is the energy of the lowest 5*d* level for RE^3+^ ions doped into compound A (*E*_5*d*_(KLaF_4_) = 52,880 cm^−1^). For Pr^3+^ in KLaF_4_, *D*(KLaF_4_) is calculated as 8700 cm^−1^, aligning with the data for other fluoride crystals [12]. Although the values of *ε_c_*, *ε_cfs_*, and *D*(A) reported in Refs. [12,14,15,16,17] were calculated for Ce^3+^ ions, Dorenbos suggests these parameters are similar for all Ln^3+^ ions when doped in the same host compound [37].

Considering the influence of the crystalline environment on the position of the Pr^3+^ 5*d* levels, fluoride matrices emerge as suitable candidates for the PCE process due to the relatively high energy of the lowest 5*d* level. Since the first allowed 4*f* → 5*d* transition in KLaF_4_:Pr^3+^ has an energy of 52,880 cm^−1^, the lowest 5*d* level must be located above the ^1^S_0_ (*E*(^1^S_0_) ≈ 47,000 cm^−1^) [38]. Figure 3b depicts the emission spectrum of KLaF_4_:1%Pr^3+^ recorded under 160 nm excitation. Seven bands were observed at 216, 237, 252, 272, 338, 405, and 484 nm. They all exhibit significantly smaller FWHM values, as indicated in Appendix A, in comparison to the bands observed in the excitation spectrum. This implies that they align with 4*f* → 4*f* transitions. The first six bands can be assigned to transitions from the ^1^S_0_ level to the ^3^H_4_, ^3^H_6_, ^3^F_3_, ^1^G_4_, ^1^D_2_, and ^1^I_6_ states, respectively, while the last band is considered to be ^3^P_0_ → ^3^H_4_. The experimental branching ratios (*β*_ex_) of the transitions from the ^1^S_0_ level were calculated as the ratio between the area under the specific peak and the area of the spectrum in the 22,500–55,000 cm^−1^ range. The obtained results are listed in Appendix A. It could be noted that the highest values were observed for the ^1^S_0_ → ^1^G_4_ and ^1^S_0_ → ^1^I_6_ transitions due to their spin-allowed character. According to Kück [39], the typical branching ratio of ^1^S_0_ → ^1^I_6_ transitions varies from 60 to 80% depending on the host type. Here, a *β*_ex_ (^1^S_0_ → ^1^I_6_) of 39.8% is much smaller than the expected value. This discrepancy is attributed to the significant influence of instrumental factors, such as the efficiency of the photomultiplier and the specifications of the diffraction grating, on the experimental branching ratio. The measured spectrum was corrected for both of these factors. However, even if correction is performed, the real relative intensities of the observed transitions can remain unknown.

A schematic illustration of the PCE process observed in KLaF_4_:Pr^3+^ under VUV excitation is presented in Figure 4. After the absorption of one 160 nm photon, Pr^3+^ ions are excited into a 4*f*5*d* configuration and then relax nonradiatively into the lower-lying ^1^S_0_ level. During radiative relaxation into the ground state, the emission of two photons occurs at 406 (^1^S_0_ → ^1^I_6_) and 484 nm (^3^P_0_ → ^3^H_4_). Because the lowest 5*d* level is located about 6000 cm^−1^ above ^1^S_0_, 5*d* → 4*f* transitions are not observed.

#### 3.2.2. Visible Excitation

As KLaF_4_ crystals doped with Pr^3+^ ions are introduced in this paper for the first time, an exploration of their optical properties in the visible range was undertaken. Figure 5a presents the excitation spectrum of KLaF_4_:0.5%Pr^3+^ monitored at 608 nm. Three 4*f* → 4*f* transitions, ^3^H_4_ → ^3^P_2_, ^3^P_1_, and ^3^P_0_, were observed at 444, 468, and 480 nm, respectively. The band at 484 nm corresponds to the transition from the first excited Stark sublevel of the ^3^H_4_ state, which is thermally populated at room temperature. Upon exciting the sample with 444 nm light, emissions from the ^3^P_1_ and ^3^P_0_ levels were detected (Figure 5b). The most intense bands observed at 484, 608, 640, and 720 nm correspond to the spin-allowed transition from ^3^P_0_ to the ^3^H_4_, ^3^H_6_, ^3^F_2_, and ^3^F_4_ levels, respectively [42].

To investigate the influence of the dopant concentration, emission spectra were acquired under 444 nm excitation for samples with varying Pr^3+^ ion concentrations but under the same measurement conditions (Appendix A). The highest relative intensity of visible luminescence was obtained for the sample doped with a 0.5% content (see inset in Figure 5b). For a higher Pr^3+^ content, concentration quenching of the luminescence occurs. The process responsible for concentration quenching is cross-relaxation (CR), which depends on the distance between the ions involved in these phenomena. As the concentration increases, the average distance between the activator ions is shortened, leading to a rise in the effectiveness of CR. The two main CR processes responsible for quenching the Pr^3+^ luminescence can be described using the following equations:{^3^P_0_, ^3^H_4_} → {^1^D_2_, ^3^H_6_}{20,809; 186} → {16,775; 4275} − 55 [cm^−1^](2)
{^1^D_2_, ^3^H_4_} → {^1^G_4_, ^3^F_3, 4_}.{16,775; 0} → {9694; 6969} + 112 [cm^−1^](3)
The energies of the Stark levels used in Equations (2) and (3) were obtained from Appendix A. Following the cross-relaxation (CR) process, there is a slight energy mismatch (as is evident in Equations (2) and (3)), which can be easily compensated by the host phonon energies. Consequently, it can be assumed that both processes are practically resonant at room temperature.

Moreover, concentration quenching significantly influences the lifetime of the emitting levels. Figure 5c,d show the emission decay curves of the ^3^P_0_ and ^1^D_2_ levels, respectively, measured for different activator ion contents. Only the sample with the lowest dopant concentration (0.1%) exhibited an exponential decay profile at both levels. When the concentration is increased, the decay patterns become increasingly non-exponential. To calculate the average observed decay time *τ_OB_*, a bellowed equation was applied [43]:(4)$$\tau_{OB}=\int_{0}^{\infty }tI\left(t\right)\;dt/\int_{0}^{\infty }I\left(t\right)\;dt$$
where *I*(*t*) and *I*(0) indicate the luminescence intensity at time *t* and *t* = 0, respectively. The obtained results are listed in Table 1. The luminescence of both levels exhibits the longest decay time for the smallest concentration of Pr^3+^ ions (37 μs for ^3^P_0_ and 346 μs for ^1^D_2_). The lifetime of the ^1^D_2_-level emissions decreases more rapidly with the concentration of Pr^3+^ than that of the ^3^P_0_ level. In the CR described using Equation (2), spin-forbidden transitions are involved, while the one occurring from level ^1^D_2_ is spin-allowed, making it more efficient.

Since the observed luminescence decay is a result of the radiative and nonradiative relaxation paths, the observed transition rate (*W_OB_*) can be described as:(5)$$W_{OB}=W_{R}+W_{NR}=\frac{1}{\tau_{OB}},$$
where *W_R_* and *W_NR_* are the rates of the radiative and nonradiative transitions, respectively. There are two main processes involved in the nonradiative relaxation of ions: multiphonon relaxation (MNR) and CR. Due to the small phonon energies of KLaF_4_ and the large energy difference between the ^3^P_0_ and ^1^D_2_ (4034 cm^−1^) and ^1^D_2_ and ^1^G_4_ levels (7081 cm^−1^), the MNR process has a neglected influence on the ^3^P_0_ and ^1^D_2_ decay times. According to this, *W_NR_* is equal to *W_CR_*, which is the rate of the cross-relaxation processes:(6)$$W_{CR}=\frac{1}{\tau_{OB}}-\frac{1}{\tau_{R}},$$
where *τ_R_* is the radiative decay time.

For the calculation of radiative lifetimes, the Judd–Ofelt theory proves invaluable. Independently proposed by Brian Judd [44] and George S. Ofelt [45] in 1962, this approach facilitates the determination of the Einstein coefficient *A_aJ,bJ_* for the spontaneous emission of electric dipole transitions, and subsequently the radiative lifetime *τ_R_* = 1/*A_aJ,bJ_*, using the formula:(7)$$A_{aJ,bJ’}=\frac{{64\pi }^{4}e^{2}}{3h\left(2J+1\right)}\frac{1}{\lambda^{3}}\frac{{n\left(n^{2}+2\right)}^{2}}{9}S\left(aJ:bJ’\right).$$

Here *e*, *h*, *λ*, and *n* represent the electron charge, Planck’s constant, transition wavelength, and refractive index, respectively. *J* denotes the total angular momentum of the initial state, while *S* signifies the line strength of the electric dipole transition between the two J (spin–orbit) multiplets *a* and *b*.

The line strength within the Judd–Ofelt theory is expressed as follows:(8)$$S\left(aJ:bJ’\right)=\sum_{t=2,\;4,\;6}\Omega_{t}{\left|\left\langle aJ\left\|U^{t}\right\|\left.bJ’\right\rangle \right.\right|}^{2}$$
with summation over all the components 2J + 1. The $\left|\left\langle aJ\left\|U^{t}\right\|\left.bJ’\right\rangle \right.\right|$ terms represent the reduced-matrix elements of the matrix, wherein the transition values between each level of a rare-earth ion are calculated, independent of the surrounding environment. These were calculated and tabulated by Carnal [46]. *Ω_λ_* (λ = 2, 4, 6) are the phenomenological parameters linked to the host. These parameters are determined using fitting by comparing the experimental oscillator strengths measured for as many transitions in the absorption spectrum as possible with theoretical ones, expressed as:(9)$$f_{ed}=\frac{8\pi^{2}mc}{3h\lambda \left(2J+1\right)}\frac{{\left(n^{2}+2\right)}^{2}}{9n}S\left(aJ:bJ’\right),$$
where *f_ed_* denotes the electric dipole oscillator strength, with the other constants as previously explained. This classical approach, although highly successful [47], is restricted to monocrystals and glasses, as it necessitates measuring the absorption spectra using transmission techniques, along with the exact dopant concentration and refractive index of the host. A thorough introduction to this theory by Robert D. Peacock is recommended for interested readers [48].

Towards the end of the 20th century, Brazilian colleagues proposed a useful method for calculating the *Ω_λ_* parameters for Eu^3+^ based on emission spectra [49,50]. They observed that for several transitions, only one $\left|\left\langle aJ\left\|U^{t}\right\|\left.bJ’\right\rangle \right.\right|$ parameter differs from zero, enabling straightforward calculation of *Ω_λ_*. This approach paved the way for further innovation, such as utilizing excitation spectra, as demonstrated by W. Luo et al. [51]; diffuse reflection spectra, as explored by Gao et al. [52]; or fluorescence decay analyses, as verified by M. Luo et al. [53]. In this study, we employed a method proposed by Ćirić et al. [54] to determine the *Ω_λ_* parameters from the emission spectra.

With the *Ω_λ_* parameters determined, it becomes feasible to calculate the Einstein coefficient of spontaneous emission, radiative decay times, emission branching ratios, and the rates of nonradiative transitions. These parameters were calculated by us using the web application published by Ćirić et al., and their values were equal to 0.488, 1.186, and 4.659 [all in 10^−20^ cm^2^] for *Ω*_2_, *Ω*_4_, and *Ω*_6_, respectively. Thus, it was possible to calculate the radiative lifetimes of the ^3^P_0_ and ^1^D_2_ levels, which are 42 and 582 μs, respectively, taking n = 1.6 as the refractive index and data from the 300 K emission spectra. In our view, this was justified for a primary reason, it ensured compliance with the assumptions of the Judd–Ofelt theory regarding the equal occupancy of the 4*f* electronic configuration levels. Therefore, using the classical method, absorption spectrum data obtained at room temperature should be utilized for calculations. In his study, Ćirić et al. suggest measurements at 77 K to avoid the presence of the ^1^D_2_ → ^3^H_4_ transition in the spectrum. However, in the case of fluoride, the lattice vibrations are minimal, and as a result, this transition is not observed at low dopant concentrations.

Since cross-relaxation is a process within a pair of ions, W_CR_ increases with increases in the Pr^3+^ concentration for both the ^3^P_0_ and ^1^D_2_ levels. The order of magnitude of the CR rate is the same for both levels when comparing samples with the same dopant content. However, it is important to note that CR rates are not absolute values. To compare the CR rates of the two processes, it is appropriate to normalize them to the rates of the radiative transitions of the levels involved. This approach allows not only the comparison of different CR processes within a matrix but also the comparison of CR processes occurring in different matrices. Figure 6 illustrates the dependence of the W_CR_/W_R_ ratio on the function of Pr^3+^ ion concentration. One can see that the effectiveness of the CR process occurring from level ^1^D_2_ is enormous compared to the CR arising from ^3^P_0_, being for the 2% sample 14 times more effective at the depopulation of the^1^D_2_ level than the radiative process.

#### 3.2.3. Thermal Behavior of Luminescence

The optical properties of KLaF_4_:Pr^3+^ were investigated at different temperatures as well. Figure 7 presents the excitation and emission spectra of the KLaF_4_:0.5%Pr^3+^ powder measured at 25 K, compared with those recorded at 300 K. Assignment of all the detected peaks is showed in both spectra. As expected, at a low temperature, the spectral lines are narrower, and their multiple-component shape can be easily observed. Based on these 25 K spectra, the experimental energies of the ^2S+1^L_j_ Stark levels were calculated and are listed in Appendix A. Please note that the energies of the ^3^F_3_, ^1^G_4_, ^1^D_2_, and ^1^S_0_ levels were estimated based on the room-temperature spectra (Figure 3b and Figure 5b) because no transitions involving those levels were observed in the low-temperature spectra.

Comparing the 25 K and 300 K excitation spectra of KLaF_4_:0.5%Pr^3+^ (Figure 7a), the band at 484 nm, which occurs at room temperature, is not observed at a low temperature. This signal is associated with a transition from the second excited crystal field level of the ground state (here named ^3^H_4_(2)) to the ^3^P_0_ level. According to Boltzmann’s statistical law, the electron distribution between the two electron levels obeys the dependence $\frac{N_{1}}{N_{0}}\;\sim \exp\left(-\frac{∆E}{k_{B}T}\right)$, where *N_1_* and *N_0_* are the populations of the higher and lower levels, respectively; *ΔE* is the energy gap between these levels; *k_B_* represents the Boltzmann constant; and *T* is the temperature. Since the *ΔE* between ^3^H_4_(0) and ^3^H_4_(2) is 147 cm^−1^ (see Appendix A), both levels are populated and participate in sample excitation at 300 K. However, when the temperature is lowered to 25 K, the thermal population of the ^3^H_4_(2) level is reduced, and only transitions from the ^3^H_4_(0) level are detected. A similar effect is observed in the low- and room-temperature emission spectra (Figure 7b) recorded under 444 nm excitation. At 300 K, emissions from both the ^3^P_0_ and ^3^P_1_ levels are observed, but at 25 K, transitions from the ^3^P_1_ level are quenched.

To better understand the thermal behaviour of Pr^3+^ luminescence in KLaF_4_ crystals, emission spectra under 444 nm excitation were recorded in the 85–760 K temperature range (Figure 8), with a 25 K interval. The observed trend reveals thermal quenching of luminescence occurs while the sample temperature is increased, which is also presented in the inset. Considering the whole measured temperature range, the *I*(*T*) function exhibits a complex nature and cannot be fitted using one simple formula. This is because integrated luminescence includes the intensities of multiple transitions, which are characterized by different thermal behaviours. Generally, to fit *I*(*T*) dependence, the Arrhenius equation is used:(10)$$I\left(T\right)=\frac{I(0)}{1+C\;exp\left(\frac{-∆E}{k_{B}T}\right)},$$
where *I*(*T*) is the luminescence intensity measured at temperature *T*, *I*(0) is the initial intensity, *C* means constant, and Δ*E* is the activation energy of the thermal quenching. In Figure 9a, the plot depicts the dependence of $\ln\left(\frac{I\left(0\right)}{I\left(T\right)}-1\right)$ on $\frac{1}{k_{B}T}$ for the ^3^P_0_ → ^3^H_4_ radiative transition. The integral intensity I(0) was derived by integrating the 480 nm multiplet in the 85 K spectrum. Linear dependence can be distinguished for temperatures ranging from 235 to 585 K. In this range, the data can be well fitted using a linear equation, where the slope is equal to Δ*E* as follows:(11)$$\ln\left(\frac{I\left(0\right)}{I\left(T\right)}-1\right)=-∆E\frac{1}{k_{B}T}+lnC.$$

The Δ*E* for the thermal quenching of ^3^P_0_ → ^3^H_4_ luminescence is determined to be 650 cm^−1^ within the temperature range of 235 K to 585 K. Notably, this value closely aligns with the energy difference between the ^3^P_0_ and ^3^P_1_ levels, calculated to be 622 cm^−1^ (see Appendix A). As a consequence, the quenching of the ^3^P_0_ emission should be explained by the thermal population of the ^3^P_1_ level when the temperature is raised from 235 to 585 K. Moreover, analyzing the thermal behaviour of the ^3^P_1_ → ^3^H_4_ transition (Figure 9b), it has been noticed that the luminescence intensity increases in the 85–510 K range, confirming that the ^3^P_1_ level is populated by thermal activation. Increasing the temperature creates an additional channel for the depopulation of the ^3^P_0_ level via CR:{^3^P_1_, ^3^H_4_} → {^1^D_2_, ^3^H_6_}  {21,352; 0} → {16,775; 4457} − 120 [cm^−1^].(12)

Recently, a distinct luminescence thermometry strategy based on the emission of praseodymium-doped phosphors has been elaborated. Among others, down-converted emissions from the two thermally coupled excited states ^3^P_0,1_ of Pr^3+^ were recorded in a temperature range of 293–593 K for YAG:Pr^3+^ [55]. This investigation revealed a remarkable temperature sensitivity, reaching up to 0.0025 K^−1^ at 573 K. Additionally, A.S. Rao observed the different temperature dependence of four various emission bands corresponding to ^3^P_0_ → ^3^H_4_, ^3^P_1_ → ^3^H_4_, ^1^D_2_ → ^3^H_4_, and ^3^P_0_ → ^3^F_2_ praseodymium transitions [56].

Consequently, four fluorescence intensity ratio models based on the relationships between different emission peaks were examined, and finally a maximum relative sensitivity was found to be 1.03% K^−1^. Previously, Pr^3+^-doped tungstate phosphors were examined as well by Ruoshan Lei et al. [57], and a similar evaluation strategy resulted in quite a high relative sensitivity (1~3.25% K^−1^) and low temperature uncertainty (0.15–0.5 K) within a wide temperature range. Another exploration focused on the emission intensity variation in ^3^P_0_ → ^3^H_4_ and ^1^D_2_ → ^3^H_4_ at lower temperatures in YPO_4_:Pr^3+^ nanopowders [58]. The obtained values for the maximum absolute and relative sensitivities were 4.60 × 10^−3^ K^−1^ at 100 K and 2.30% K^−1^ at 10 K, respectively. Additionally, glass materials have been employed in the development of luminescence thermometers. Notably, Maturi et al. recently presented Yb^3+^/Pr^3+^ co-doped fluoride phosphate glasses working as primary thermometers, demonstrating a relative thermal sensitivity and uncertainty of 1.0% K^−1^ and 0.5 K, respectively [59].

In relation, we observed the different effect of temperatures of 80–750 K on several emission bands of praseodymium originating in the ^3^P_0_ and ^3^P_1_ multiplets. We utilized the fluorescence intensity ratio (FIR) expressed using the equation:(13)$$FIR=\frac{I_{3P1}}{I_{3P_{0}}}=B\exp\left(-\frac{\Delta E}{k_{B}T}\right)$$
where Δ*E* is the energy gap between the two thermally coupled levels and *B* is the temperature-independent constant. Figure 10 displays the effect of temperature (80–750 K) on the fluorescence intensity ratio corresponding to certain praseodymium emission bands. A reliable fit in applying Equation (13) was achieved for Δ*E* = 567 cm^−1^. It follows from these plots that the fluorescence intensity ratios attributed to the thermally coupled levels ^3^P_0_ and ^3^P_1_ rise with an increasing temperature, reaching their highest values at 750 K. The variations in the absolute and relative FIR changes *S_A_* and *S_R_* with temperature are expressed as [60]:(14)$$S_{A}=\frac{dFIR}{dT}=FIR\frac{\Delta E}{kT^{2}}$$
and
(15)$$S_{R}=\frac{1}{FIR}\frac{dFIR}{dT}\cdot 100\%=\frac{\Delta E}{kT^{2}}\cdot 100\%$$

The potential application of optical material as a luminescence thermometer can be assessed utilizing these *S_A_* and *S_R_* parameters, which determine the thermosensitive phosphor properties. In the case of Pr^3+^-doped KLaF_4_, the most efficient relative temperature sensitivities were found to be 1.70% K^−1^ at T = 140 K and 1.45% K^−1^ at T = 175 K for the (^3^P_1_ → ^3^H_4_/^3^P_0_ → ^3^H_4_) and (^3^P_1_ → ^3^H_5_/^3^P_0_ → ^3^H_5_) transitions, respectively. These values are compared with those reported for other Ln^3+^-based luminescence thermometers in Appendix A [61,62,63,64,65,66,67,68].

The temperature resolution δT (uncertainty in temperature), which refers to the minimal temperature change that causes significant fluctuation in an examined parameter, was estimated according to the formula [60]:(16)$$\delta T=\frac{1}{S_{R}}\frac{\delta ∆}{∆}$$
where *δ*Δ/Δ represents the relative error in determining the thermometric parameters (around 0.5% for emission ratiometric measurements [69]). For the (^3^P_1_ → ^3^H_4_/^3^P_0_ → ^3^H_4_) and (^3^P_1_ → ^3^H_5_/^3^P_0_ → ^3^H_5_) ratiometric estimations, the relative sensitivity with temperature resolutions of 0.29 K and 0.34 K over the whole measurement range can be estimated, respectively. Eventually, the thermographic qualities of the studied phosphor are especially promising for potential optical sensing considering the 125–250 K temperature region.

## 4. Conclusions

KLaF_4_ crystalline powders doped with different Pr^3+^ ion contents were synthesized for the first time, and their optical properties were investigated. Although all papers dealing with KLaF_4_ crystals have previously reported a cubic or hexagonal structure, here, an orthorhombic *Pnma* space group was observed. The obtained crystallites, which were synthesized via a high-temperature solid-state reaction, had an agglomerated, non-uniform shape with a main grain size of 6.9 μm.

The optical properties of the prepared phosphor were investigated in the UV and visible ranges. The influence of the host crystal structure on the position of the Pr^3+^ 5*d* levels was analyzed according to the UVC excitation spectrum. It was found that the energy of the lowest 5*d* level (52,880 cm^−1^) is higher than the energy of the ^1^S_0_ level (46,387 cm^−1^), meaning the PCE process should be observed for this system. The occurrence of the quantum-cutting effect was proven through observation of the ^1^S_0_ → ^3^I_6_ (406 nm) and ^3^P_0_ → ^3^H_4_ (484 nm) transitions upon exciting the sample with 160 nm radiation. Under blue light excitation (444 nm), emissions from the ^3^P_0_ and ^3^P_1_ levels were observed at 300 K. In contrast, the 25 K emission spectrum contained only transitions from the ^3^P_0_ level. The Arrhenius plot calculation revealed that the thermal population of the ^3^P_1_ level is responsible for the difference between the room- and low-temperature spectra. Furthermore, the concentration-quenching effect was observed as a result of the increasing cross-relaxation rate for both the ^3^P_0_ and ^1^D_2_ levels. However, analysis of the cross-relaxation rates revealed that the CR process is much more effective for the ^1^D_2_ level. Finally, the thermometric properties of the prepared system were investigated based on transitions involving the thermally coupled levels ^3^P_0_ and ^3^P_1_. The most efficient relative temperature sensitivity was found to be 1.70% K^−1^ at T = 140 K.

Summing up, the newly synthesized material KLaF_4_:Pr^3+^ was found to be an effective PCE phosphor, which makes it attractive for various optical applications.

## Figures and Tables

**Figure 1 materials-17-01410-f001:**
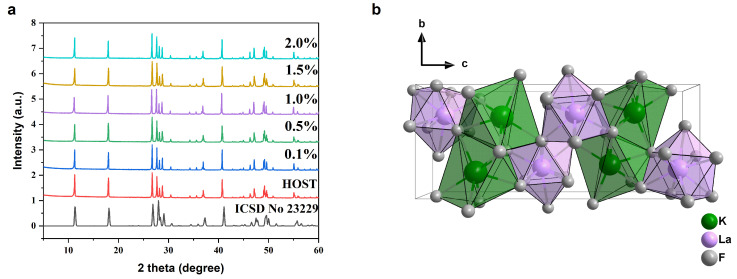
(**a**) X-ray powder diffraction patterns of KLaF_4_:Pr^3+^ particles and the orthorhombic KCeF_4_ standard data (ICSD file No. 23229) [34]. (**b**) Crystal structure of KLaF_4_ (orthorhombic system with the *Pnma* space group) along the *a* axis.

**Figure 2 materials-17-01410-f002:**
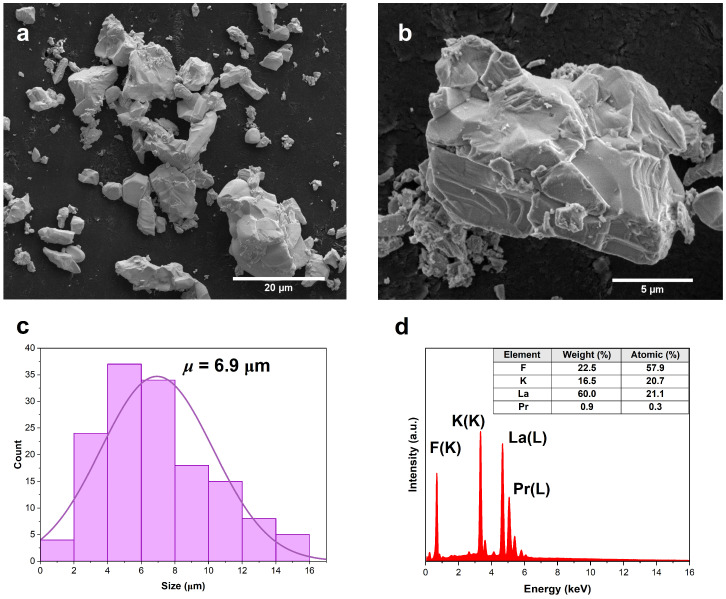
(**a**,**b**) SEM images of KLaF_4_ grains at two different scales. (**c**) Particle size distribution histogram. The purple line refers to the normal distribution function with *μ* = 6.9 μm and *σ* = 3.3 μm. (**d**) EDS spectrum of the KLaF_4_:1.5%Pr^3+^ sample; inset shows weight and atomic percentages of elements in the matrix.

**Figure 3 materials-17-01410-f003:**
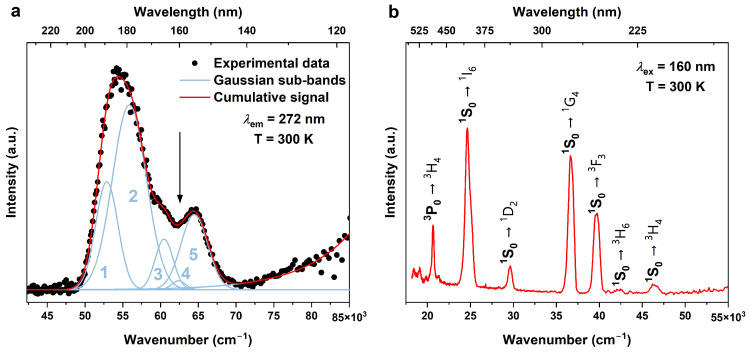
(**a**) Excitation spectrum of the KLaF_4_:1%Pr^3+^ sample monitored at 272 nm. The arrow indicates the excitation energy used for emission measurement and the numbers 1–5 refer to peak numbers listed in Appendix A (**b**) Emission spectrum of the KLaF_4_:1%Pr^3+^ sample measured under 160 nm excitation. Both spectra were recorded at T = 300 K.

**Figure 4 materials-17-01410-f004:**
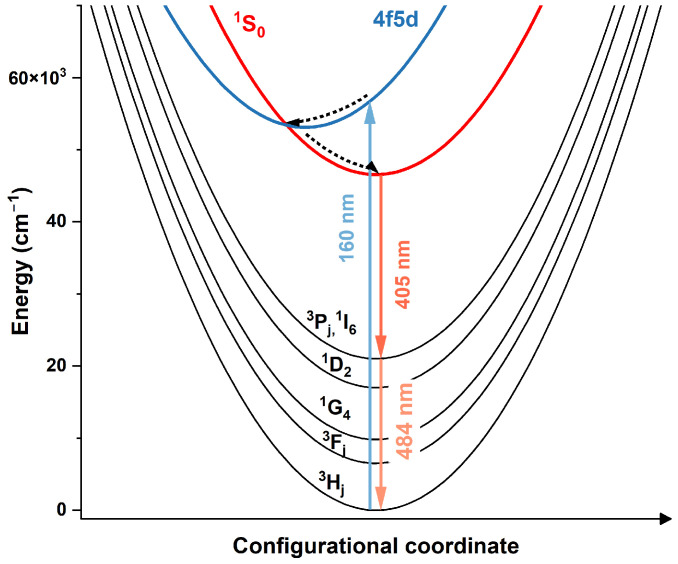
Schematic illustration of the PCE process in KLaF_4_ doped with Pr^3+^ ions. Solid arrows represent radiative transitions and dashed arrows correspond to nonradiative ones. Other observed transitions from the ^1^S_0_ level were not included in the graph. Please note that according to the work of Seijo et al. [40]. and Mahlik et al. [41], the metal–ligand (ML) distance is smaller for the 5*d* electronic configuration than for the 4*f*.

**Figure 5 materials-17-01410-f005:**
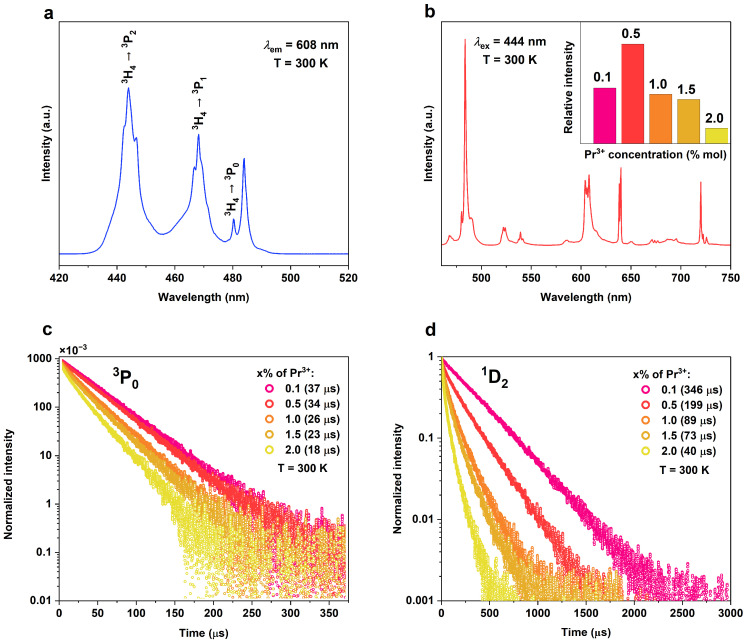
(**a**) Excitation (*λ_em_* = 608 nm) and (**b**) emission (*λ_ex_* = 444 nm) spectra of KLaF_4_:0.5%Pr^3+^ measured at 300 K. The inset presents the relative luminescence intensity for different Pr^3+^ ion concentrations. Decay time curves of the (**c**) ^3^P_0_ and (**d**) ^1^D_2_ levels recorded at 300 K for samples with different Pr^3+^ contents.

**Figure 6 materials-17-01410-f006:**
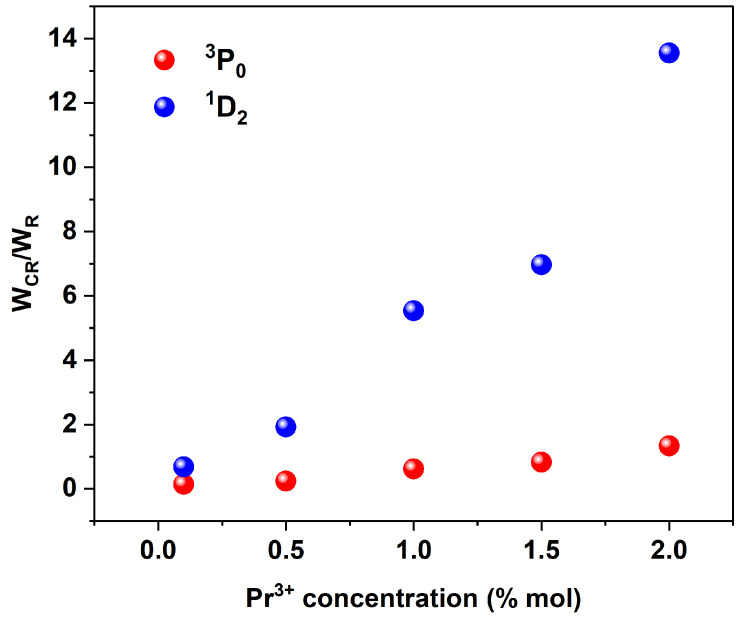
Cross-relaxation rates (*W_CR_*) normalized to the radiation transition rate (*W_R_*) of ^3^P_0_ (red) and ^1^D_2_ (blue) levels for different Pr^3+^ ion concentrations.

**Figure 7 materials-17-01410-f007:**
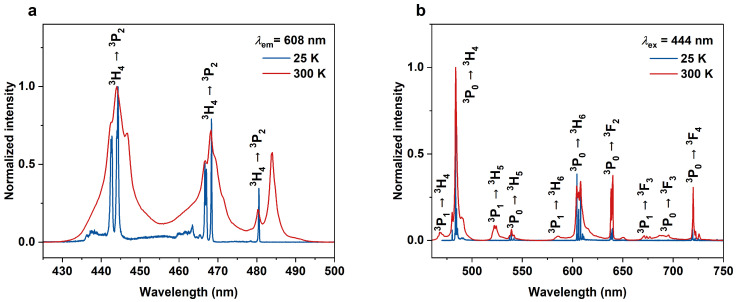
(**a**) Excitation (*λ*_em_ = 608 nm) and (**b**) emission (*λ*_ex_ = 444 nm) spectra of KLaF_4_:0.5%Pr^3+^ measured at 25 K (blue lines) and 300 K (red lines).

**Figure 8 materials-17-01410-f008:**
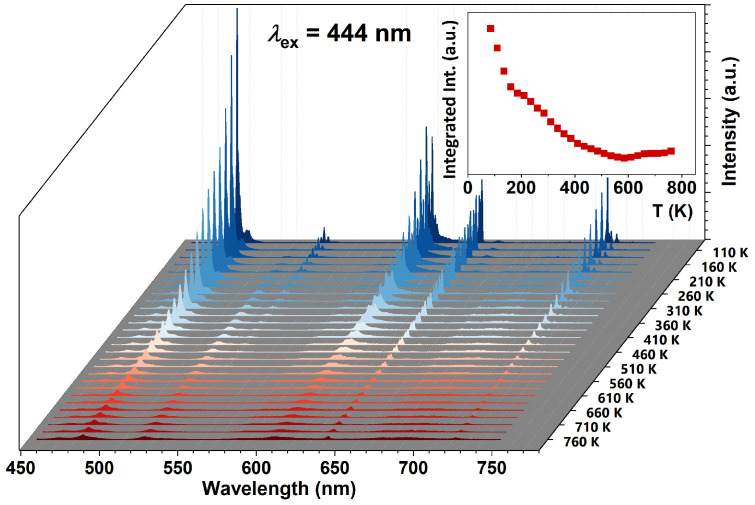
Emission spectra of the KLaF_4_:0.5%Pr^3+^ sample measured in the 85–760 K temperature range under 444 nm excitation. In the insets, the dependence of integrated luminescence intensity on temperature was plotted.

**Figure 9 materials-17-01410-f009:**
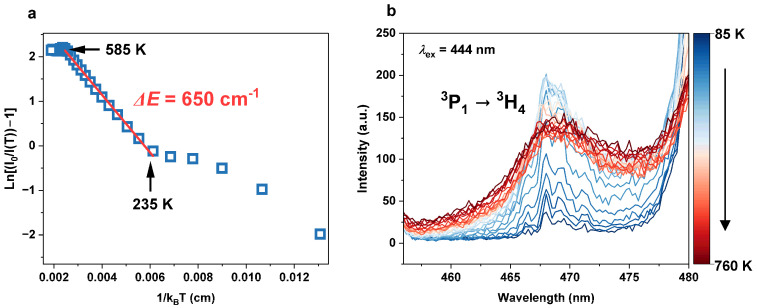
(**a**) Arrhenius plot for calculating activation energy of thermal quenching of ^3^P_0_ → ^3^H_4_ transition in KLaF_4_:0.5%Pr^3+^. (**b**) Emission band corresponding to ^3^P_1_ → ^3^H_4_ transition measured at different temperatures from 85 to 760 K.

**Figure 10 materials-17-01410-f010:**
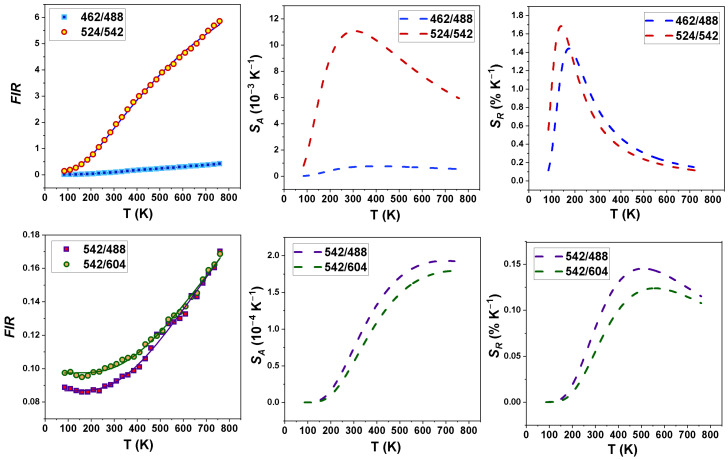
Fluorescence intensity ratios, absolute and relative sensitivity determined for (^3^P_1_ → ^3^H_4_/^3^P_0_ → ^3^H_4_); (^3^P_1_ → ^3^H_5_/^3^P_0_ → ^3^H_5_); (^3^P_0_ → ^3^H_5_/^3^P_0_ → ^3^H_4_); and (^3^P_0_ → ^3^H_5_/^3^P_0_ → ^3^H_6_) band pairs in KLaF_4_:0.5%Pr^3+^.

**Table 1 materials-17-01410-t001:** Radiative decay times (*τ_R_*), observed decay times (*τ_OB_*), and cross-relaxation rates (*W_CR_*) of the ^3^P_0_ and ^1^D_2_ levels for samples with different Pr^3+^ concentrations.

Pr^3+^ Concentration (% mol)	^3^P_0_			^1^D_2_		
	*τ_R_* [μs]	*τ_OB_* [μs]	*W_CR_* [s^−1^]	*τ_R_* [μs]	*τ_OB_* [μs]	*W_CR_* [s^−1^]
0.1	42	37	3.2 × 10^3^	582	346	1.2 × 10^3^
0.5		34	5.6 × 10^3^		199	3.3 × 10^3^
1.0		26	1.5 × 10^4^		89	9.5 × 10^3^
1.5		23	2.0 × 10^4^		73	1.2 × 10^4^
2.0		18	3.2 × 10^4^		40	2.3 × 10^4^

## Data Availability

The data presented in this study are openly available in Zenodo at https://doi.org/10.5281/zenodo.10559692.

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
