# Peer review of "Luminescence Properties of an Orthorhombic KLaF4 Phosphor Doped with Pr3+ Ions under Vacuum Ultraviolet and Visible Excitation"

_materials, 2024, doi:10.3390/ma17061410_

Round 1
Reviewer 1 Report
Comments and Suggestions for Authors
In this manuscript, the authors presented the optical spectroscopic properties of Pr3+ doped orthorhombic-phased KLaF4 phosphor derived from a solid-state reaction in an H2-reducing atmosphere. The obtained results are novel and interesting. Therefore, I recommend accepting it for publication after proper revisions.
1/ The manuscript title should be re-phrased. The orthorhombic structure and the polycrystalline powder (or phosphor) other than single-crystal should also reflected in the title.
2/ The abbreviations in this manuscript were defined in a mess, for example, praseodymium was defined as both Pr and Pr3+, the symbol ‘’ was incorrectly used in abbreviation definition, X-ray powder diffraction was defined as XRD, in fact, it should be XRPD. There are too many incorrect definitions. Moreover, some abbreviations were not defined, for example, UV and EDS.
3/ The first letters of some words in the sentences are not necessary to be capitalized, for example, Quantum Efficiency, Reduced-Matrix Elements and Fluorescence Spectrometer.
4/ In section 3.2.1, the authors told too much things which are not derived from the present study but concluded from published papers. My suggestion is this section should be shortened and some necessary references should be maintained.
5/ In Fig. 5(b) each peak should be assigned to its corresponding transition.
6/ To calculate the decay time, Eq. (4) for the single exponential decay is no problem, but for the non-exponential decay Eq. (4) does not fit. Obviously, in Fig. 5 (d) some decays for the samples with high Pr3+ concentrations follow non-exponential decay. Therefore, Eq. (4) should be corrected. The authors should read some papers to correct this equation.
7/ The Judd-Ofelt parameters of the studied samples were calculated via the approach in Ref.[48] that is not the traditional Judd-Ofelt calculation procedure. However, the authors just presented some basic formulas of Judd-Ofelt theory, the specific calculation process was ignored. My suggestion is that the authors should state some progress in the Judd-Ofelt calculation strategy, and meanwhile briefly state the calculation process in this work. Except for the calculation approach in Ref.[48], The Judd-Ofelt calculation could also be carried out using excitation spectrum [Phys. Chem. Chem. Phys. 12(2010)3276], diffuse reflection spectrum [Light: Science & Applications, 13(2024)17], and fluorescence decays [Phys. Chem. Chem. Phys. 22(2020)25177]. This progress of the Judd-Ofelt calculation should be mentioned in the manuscript.
8/ Another question is whether the Judd-Ofelt parameters depend on the Pr3+ doping concentration.
9/ In the figure caption for Fig. 10, both the symbols “-” and “->” are used in the expression for the optical transitions. My suggestion is to use an identical symbol.
10/ Some relevant papers published in the last 3 years should also be cited in this manuscript, the authors should find these papers, read them, comment on them in the text, and list them in the references.
11/ In general, written English is OK for publication, but there are still some bad sentences that can be difficult to understand. Therefore, the authors should further polish their English.
Comments on the Quality of English Language
In general, written English is OK for publication, but there are still some bad sentences that can be difficult to understand. Therefore, the authors should further polish their English.
Author Response
Wrocław February 8, 2024
Reviewers' comments and remarks regarding the article: “Luminescence properties of KLaF4 doped with Pr3+ ions under VUV and visible excitation” and the authors' answers.
We would like to thank all reviewers for their insightful work. We believe that a good article is the result of the joint work of authors and reviewers. It is thanks to their comments that the article could be improved.
All reviewer comments were addressed. Changes we made to the manuscript are marked in yellow highlighted text.
Best Regards,
- Deren on behalf of all authors
List of changes and rebuttals:
Responses to Reviewer 1:
Reviewer's comments:
In this manuscript, the authors presented the optical spectroscopic properties of Pr3+ doped orthorhombic-phased KLaF4 phosphor derived from a solid-state reaction in an H2-reducing atmosphere. The obtained results are novel and interesting. Therefore, I recommend accepting it for publication after proper revisions.
1/ The manuscript title should be re-phrased. The orthorhombic structure and the polycrystalline powder (or phosphor) other than single-crystal should also reflected in the title.
Response to Reviewer 1, comment No. 1:
Thank you for this comment. We re-phrased the title for:
“Luminescence properties of orthorhombic KLaF4 phosphor doped with Pr3+ ions under vacuum ultraviolet and visible excitation”.
2/ The abbreviations in this manuscript were defined in a mess, for example, praseodymium was defined as both Pr and Pr3+, the symbol ‘’ was incorrectly used in abbreviation definition, X-ray powder diffraction was defined as XRD, in fact, it should be XRPD. There are too many incorrect definitions. Moreover, some abbreviations were not defined, for example, UV and EDS.
Response to Reviewer 1, comment No. 2:
Thank you for your comment. We have addressed all the identified mistakes. In the text, we exclusively designate praseodymium as 'Pr' once (line 28), where we refer to it as an element. In all other cases, we specifically discuss praseodymium ions, which we denote as 'Pr3+'.
3/ The first letters of some words in the sentences are not necessary to be capitalized, for example, Quantum Efficiency, Reduced-Matrix Elements and Fluorescence Spectrometer.
Response to Reviewer 1, comment No. 3:
We accept the comment. All mistakes were corrected.
4/ In section 3.2.1, the authors told too much things which are not derived from the present study but concluded from published papers. My suggestion is this section should be shortened and some necessary references should be maintained.
Response to Reviewer 1, comment No. 4:
Thank you for this valuable comment. In this part of the manuscript, we intend to explain the aspects that determine the position of the lowest 5d level and make KLaF4 polycrystals a favorable host for photon cascade emitters. However, we agreed that such a long discussion was not necessary and shortened it as follows:
“The 5d-level positions of lanthanide ions are influenced by factors such as centroid shift (εc), crystal field splitting (εcfs), and redshift (D(A))[14]. Dorenbos investigated these parameters across various crystalline lattices, including fluorides [14], chlorides [15], oxides [16], and aluminates [17].
The centroid shift (εc) represents the energy difference between the average positions of 5d levels in a free RE3+ ion and within a crystalline host. The εc value is influenced by the coordinating ligand and is the smallest for the fluoride matrices according to the nephelauxetic series:
F- < Cl- < Br- < I- < O2- < S2-.
Crystal field splitting (εcfs) is the energy difference between the lowest and highest 5d components. It tends to increase with decreasing coordination number. In the orthorhombic KLaF4 lattice, La3+ ions are coordinated by 9 fluorine anions, resulting in small εcfs (11,592 cm-1). Both εc and εcfs contribute to the redshift (D(A)) of the first allowed 4f → 5d transition in host A. This redshift can be expressed as:
(1)
where E5d(free) is the position of the lowest 5d level of RE3+ as a free ion (for Pr3+, E5d(free) = 61,580 cm-1), and E5d(A) likewise for RE3+ ion doped into compound A (E5d(KLaF4) = 52,880 cm-1). For Pr3+ in KLaF4, D(KLaF4) is calculated as 8700 cm-1, aligning with data for other fluoride crystals [12]. Although the values of εc, εcfs, and D(A) reported in Ref. [12,14-17] were calculated for Ce3+ ions, Dorenbos suggests these parameters are similar for all Ln3+ ions when doped in the same host compound [37].”
5/ In Fig. 5(b) each peak should be assigned to its corresponding transition.
Response to Reviewer 1, comment No. 5:
Due to the small dimension of this figure, we did not assign the observed peaks to the corresponding transitions to keep the spectrum clear and transparent. The assignment of all observed peaks was included in Figure 7(b), which contains the same spectrum (red line). However, we are aware that it may not be well understood, so we decided to add the highlighted text:
“After excitation of the sample with the 444 nm light, emission from the 3P1 and 3P0 levels was detected (Figure 5b). The most intense bands observed at 484, 608, 640, and 720 nm, correspond with the spin allowed transition from 3P0 to 3H4, 3H6, 3F2, and 3F4 levels respectively [42]. The assignment of all detected peaks is presented in Figure 7b.”
6/ To calculate the decay time, Eq. (4) for the single exponential decay is no problem, but for the non-exponential decay Eq. (4) does not fit. Obviously, in Fig. 5 (d) some decays for the samples with high Pr3+ concentrations follow non-exponential decay. Therefore, Eq. (4) should be corrected. The authors should read some papers to correct this equation.
Response to Reviewer 1, comment No. 6:
We agree that the formula in Equation (4) was written incorrectly; however, the calculations were performed according to the right formula:
,
Which is commonly used for calculating the average lifetime for non-exponential decays. In the article, we cited the work of Zatryb and Klak (Ref. [43] in the revised manuscript), which discuss the proper way for determining the decay time in different systems. Moreover, more examples of utilizing the above formula for non-exponential kinetics can be found in the literature:
- Niedźwiedzki, T.; Komar, J.; Głowacki, M.; Berkowski, M.; Ryba-Romanowski, W. Luminescence and energy transfer phenomena in Gd3 (Al,Ga)5O12 crystals single doped with thulium and co-doped with thulium and holmium. Journal of Luminescence 2017, 192, 77–84.
- Nakazawa, E. Fundamentals of Luminescence. In Phosphor handbook; Yamamoto, H.; Yen, W. M.; Shionoya, S., Eds.; CRC PRESS: Boca Raton, 2024.
- Devi, L. L.; Jayasankar, C. K. Novel reddish-orange color emitting Ca2SiO4:Sm3+ phosphors for white led applications prepared by using agricultural waste. Journal of Luminescence 2020, 221, 116996.
- Inokuti, M.; Hirayama, F. Influence of energy transfer by the exchange mechanism on donor luminescence. The Journal of Chemical Physics 1965, 43, 1978–1989.
We changed the formula in Equation (4) to the right one.
7/ The Judd-Ofelt parameters of the studied samples were calculated via the approach in Ref.[48] that is not the traditional Judd-Ofelt calculation procedure. However, the authors just presented some basic formulas of Judd-Ofelt theory, the specific calculation process was ignored. My suggestion is that the authors should state some progress in the Judd-Ofelt calculation strategy, and meanwhile briefly state the calculation process in this work. Except for the calculation approach in Ref.[48], The Judd-Ofelt calculation could also be carried out using excitation spectrum [Phys. Chem. Chem. Phys. 12(2010)3276], diffuse reflection spectrum [Light: Science & Applications, 13(2024)17], and fluorescence decays [Phys. Chem. Chem. Phys. 22(2020)25177]. This progress of the Judd-Ofelt calculation should be mentioned in the manuscript.
Response to Reviewer 1, comment No. 7:
Thank you for bringing up the Judd-Ofelt approach, which indeed holds significant utility for researchers interested in rare earth ion spectroscopy. We have expanded this aspect of our manuscript with the following paragraph:
“For the calculation of radiative lifetimes, the Judd-Ofelt theory proves invaluable. Independently proposed by Brian Judd [44] and George S. Ofelt [45] in 1962, this approach facilitates the determination of Einstein coefficient AaJ,bJ for spontaneous emission of electric dipole transitions, and subsequently the radiative lifetime τR = 1/AaJ,bJ, through the formula:
(7)
Here e, h, λ, and n represent the electron charge, Planck’s constant, transition wavelength, and refractive index, respectively. J denotes the total angular momentum of the initial state, while S signifies the line strength of the electric dipole transition between two J (spin-orbit) multiplets a and b.
The line strength within the Judd-Ofelt theory is expressed by:
(8)
with summation over all components 2J+1. The terms represent Reduced-Matrix Elements of the matrix wherein the transition values between each level of a rare earth ion are calculated, independent of the surrounding environment. These were calculated and tabulated Carnal [46]. The Ωλ (λ = 2, 4, 6) are phenomenological parameters linked to the host. These parameters are determined through fitting by comparing experimental oscillator strengths measured for as many transitions in the absorption spectrum as possible with theoretical ones, expressed as:
, (9)
where fed denotes the electric dipole oscillator strength, with the other constants as previously explained. This classical approach, although highly successful [47] is restricted to monocrystals and glasses, as it necessitates absorption spectra measured through transmission techniques, alongside exact dopant concentration and refractive index of the host. A thorough introduction to this theory by Robert D. Peacock is recommended for interested readers [48].
Towards the end of the 20th century, Brazilian colleagues proposed a useful method for calculating the Ωλ parameters for Eu3+ based on emission spectra [49, 50]. They observed that for several transitions, only one Ωλ parameter differs from zero, enabling its straightforward calculation. This approach paved the way for further innovation, such as utilizing excitation spectra as demonstrated by W. Luo et al. [51] diffuse reflection spectra explored by Gao et al. [52], or fluorescence decay analyses as verified by M. Luo et al. [ 53] In this study, we employed a method proposed by Ćirić et al. [54]. for determining the Ωλ parameters from emission spectra.
With the Ωλ parameters determined, it becomes feasible to calculate Einstein coefficient of spontaneous emission, radiative decay times, emission branching ratios, and rates of nonradiative transitions. These parameters calculated by us using the web application published by Ćirić et al. and their values were equal to 0.488, 1.186, and 4.659 [all in 10-20 cm2] for Ω2, Ω4, and Ω6 respectively. Thus, it was possible to calculate the radiative lifetimes of the 3P0 and 1D2 levels, which are 42 and 582 μs, respectively, taking n = 1.6 as refractive index and data from the 300 K emission spectra. In our view, it was justified for two primary reasons: firstly, it ensured compliance with the assumptions of the Judd-Ofelt theory regarding the equal occupancy of the 4f electron configuration levels. Consequently, in the classical method, absorption spectrum data obtained at room temperature should be utilized for calculations. In his study, Ćirić et al suggests measurements at 77 K to avoid the presence of the 1D2 → 3H4 transition in the spectrum. However, in the case of fluoride, lattice vibrations are minimal, and as a result, this transition is not observed at low dopant concentrations.”
8/ Another question is whether the Judd-Ofelt parameters depend on the Pr3+ doping concentration.
Response to Reviewer 1, comment No. 8:
Thank you for the remark. In classical Judd-Ofelt theory, phenomenological Ωλ parameters are expected to be independent of dopant concentration; any observed dependence implies a change in the matrix. Ciric et al. [1] demonstrated such observations in their studies on Y2O3 and YVO4, where they investigated the influence of temperature and Eu3+ dopant concentration on Judd-Ofelt parameters. The results obtained in his study align with intuitive expectations. Increasing temperature alters the occupation of ground and excited states, such as through thermalization of the 5D1 level with the 5D0 one. Additionally, the larger size of the Eu3+ ion compared to Y3+ suggests that an increase in dopant concentration would likely affect the unit cell's volume.
Moreover, a more pronounced effect of concentration on Ωλ parameters arises when trivalent lanthanide ions replace divalent ions in a crystal host. This substitution introduces defects in the matrix due to the need to balance charges, thereby impacting the Ωλ parameters. For instance, G. Deng et al. [2] investigated PbF2 doped with Pr3+, highlighting similar effects. In the same way, glass matrices doped with lanthanide ions are expected to exhibit changes in these parameters with increasing dopant concentration [3].
- Ćirić, A.; Stojadinović, S.; Dramićanin, M.D. Temperature and Concentration Dependent Judd-Ofelt Analysis of Y2O3:Eu3+ and YVO4:Eu3+. Physica B Condens. Matter 2020, 579, 411891; DOI:10.1016/j.physb.2019.411891.
- Deng, G.; Zhang, Y.; Xu, M.; Tao, S.; Li, S.; Fang, Q.; Zhao, C.; Hang, Y. Spectroscopic Properties of Pr:PbF2 Crystal with Different Doping Concentration. J. Lumin. 2024, 267, 120357; DOI:10.1016/j.jlumin.2023.120357.
- Tao, C.; Wu, Z.; Li, B.; Huang, F.; Tian, Y.; Lei, R.; Xu, S. Luminescence Properties of Highly Er3+-Doped Fluorotellurite Glass. Opt. Mater. (Amst) 2024, 148, 114966; DOI:10.1016/j.optmat.2024.114966.
We acknowledge the issue; however, after thorough deliberation, we have decided not to include the aforementioned paragraph in the manuscript. This decision stems from our lack of examination regarding the dependence of Omega parameters on dopant concentration and also to maintain conciseness throughout the text.
9/ In the figure caption for Fig. 10, both the symbols “-” and “->” are used in the expression for the optical transitions. My suggestion is to use an identical symbol.
Response to Reviewer 1, comment No. 9:
Thank you for the comment. The oversight was corrected.
10/ Some relevant papers published in the last 3 years should also be cited in this manuscript, the authors should find these papers, read them, comment on them in the text, and list them in the references.
Response to Reviewer 1, comment No. 10:
According to the Reviewer’s suggestion we extended the references list and cited three relevant papers:
- “1. Introduction”section:“Recently, Deo et. al. reported KLaF4 nanopartices co-doped with Eu3+, Er3+, and Yb3+ ions, which can be simultaneously excited in the visible and NIR regions, giving upconversion and downconversion emission at the same time [4]. This dual-mode approach offers useful technology for bioimaging or information encryption. Moreover, Nd3+-doped KLaF4 nanoparticle colloidal solutions were proposed as an NIR high-power liquid laser material and amplifier [5]”
- “3.2.3 Thermal behavior of luminescence” section:”Additionally, glass materials have been employed in the development of luminescence thermometers. Notably, Maturi et al. recently presented Yb3+/Pr3+ co-doped fluoride phosphate glasses working as an primary thermometer, demonstrating a relative thermal sensitivity and uncertainty of 1.0% K-1 and 0.5 K, respectively [52].”
- Deo, I.S.; Gupta, M.; Prakash, G.V. Up- and Downconversion Dual-Mode Excitation Spectral Studies of Rare Earth Doped KLaF4 Nano Emitters for Biophotonic Applications. J. Phys. Chem. C 2023, 127, 24233–24241, doi:10.1021/acs.jpcc.3c05125.
- Gupta, M.; Deo, I.S.; Nagarajan, R.; Prakash, G.V. Highly Efficient Hexagonal-Phase Nd3+ Doped KLaF4 Nanoparticles Colloidal Suspension for Liquid Lasers. Opt Mater (Amst) 2022, 133, 113045, doi:10.1016/j.optmat.2022.113045.
- Maturi, F.E.; Gaddam, A.; Brites, C.D.S.; Souza, J.M.M.; Eckert, H.; Ribeiro, S.J.L.; Carlos, L.D.; Manzani, D. Extending the Palette of Luminescent Primary Thermometers: Yb3+/Pr3+ Co-Doped Fluoride Phosphate Glasses. Chem. Mater. 2023, 35, 7229–7238, doi:10.1021/acs.chemmater.3c01508.
11/ In general, written English is OK for publication, but there are still some bad sentences that can be difficult to understand. Therefore, the authors should further polish their English.
Response to Reviewer 1, comment No. 11:
We made an effort to refine the language in the work, aiming to enhance its clarity and conciseness.

Reviewer 2 Report
Comments and Suggestions for Authors
In this manuscript, the authors reported the luminescence properties of a series of KLaF4: Pr3+ with different Pr3+ concentrations. The synthesis method of KLaF4: Pr3+ materials was reported. the obtained materials were characterized by PXRD, SEM, and EDS. The authors also studied the temperature-dependent optical properties of the materials at different temperatures. The thermometric performance of the materials was also explored in the manuscript. The work is interesting; thus, I think it can be published in Materials after solving some minor problems.
1. In Figure 1a, the PXRD pattern of KLaF4: 0.5%Pr3+ is right-shifted, please correct it.
2. The EDS spectrum of at least one KLaF4: Pr3+ material should be added, as well as the element mapping of the material. these help in realizing whether the Pr3+ ions are homogeneously distributed in the final materials.
Author Response
Wrocław February 8, 2024
Reviewers' comments and remarks regarding the article: “Luminescence properties of KLaF4 doped with Pr3+ ions under VUV and visible excitation” and the authors' answers.
We would like to thank all reviewers for their insightful work. We believe that a good article is the result of the joint work of authors and reviewers. It is thanks to their comments that the article could be improved.
All reviewer comments were addressed. Changes we made to the manuscript are marked in yellow highlighted text.
Best Regards,
- Deren on behalf of all authors
Responses to Reviewer 2:
Reviewer's comments
In this manuscript, the authors reported the luminescence properties of a series of KLaF4: Pr3+ with different Pr3+ concentrations. The synthesis method of KLaF4: Pr3+ materials was reported. the obtained materials were characterized by PXRD, SEM, and EDS. The authors also studied the temperature-dependent optical properties of the materials at different temperatures. The thermometric performance of the materials was also explored in the manuscript. The work is interesting; thus, I think it can be published in Materials after solving some minor problems.
- In Figure 1a, the PXRD pattern of KLaF4: 0.5%Pr3+ is right-shifted, please correct it.
Response to Reviewer 2, comment No. 1:
Thank you for the comment. We corrected the position of the powder XRD of the KLaF4:0.5%Pr3+ sample in Figure 1a.
- The EDS spectrum of at least one KLaF4: Pr3+ material should be added, as well as the element mapping of the material. these help in realizing whether the Pr3+ ions are homogeneously distributed in the final materials.
Response to Reviewer 2, comment No. 2:
Encouraged by your comment we performed EDS analysis and element mapping of the KLaF4:1.5%Pr3+ sample and replaced the EDS spectrum in Figure 2d for spectrum of activated sample. The results of EDS mapping present homogenous distribution of all elements in the analyzed grain. We discussed these results in the “3.1. Structure” as follows:
“Figure 2d shows the energy dispersive spectroscopy (EDS) spectrum for a KLaF4:1.5%Pr3+ sample. Emission peaks were observed at 0.8 and 4.7 keV for lanthanum, 3.3 keV for potassium, 0.7 keV for fluorine, and 5.1 eV for praseodymium.(…) Furthermore, EDS mapping analysis confirmed the homogeneous distribution of K, La, F, and Pr elements (refer to Figure S1 in the Supplementary Materials).”

Reviewer 3 Report
Comments and Suggestions for Authors
The article reports the introduction of praseodymium in KLaF4 crystals. The material was characterized concerning the photoluminescence properties and the application as thermometer.
The fact that is the first report of this material makes the work with potential to be published.
But I only can agree with the publication after addressing some questions.
These are the main questions:
“The wide band beginning at around 71,000 cm-1 can be attributed to host absorption.” The spectra only show what may be the beginning of the band. Did the authors performed emission spectra excited at a wavelength between 120 and 130 nm, to be sure that the wide band is not an artifact?
Line 231: “The highest integral intensity of visible luminescence was obtained for the sample doped with the 0.5% of activator ions” Emission spectra is not a quantitative technique, the intensity of different samples can not be compared.
How the relative sensitivity values compare with other values reported for thermometers based on lanthanide ions?
Can the authors calculate the temperature uncertainty? This would be important to determine the operation range of the thermometer.
Did the authors measure the emission quantum yield?
Other minor questions:
In line 146 “photoluminescence spectrum” should be replace by “emission spectrum”, considering that an excitation spectrum is also a photoluminescence spectrum.
“VUV” is used without being defined.
Characters in some figures are too small.
In line: “The arrow indicates the excitation energy.” I understand that the wavelength was used for measured the spectrum in Figure 3b. But I do not see any advantage of having the arrow in the spectra, since that excitation wavelength is not relevant for the spectrum in Figure 3a.
Why the author chose 160 nm for the emission spectra if that is not the maximum of the excitation spectrum?
“All of them are characterized by small values of FWHM (Full Width at Half Maximum) (see Table S2)”. They are small when compare with what?
Reference is needed at the end of line 225.
That was the temperature used to measure the data in Figure 3 and 5?
There is an error in the line 17 of the supplementary material.
Reference is needed at the end of line 398.
Author Response
Wrocław February 8, 2024
Reviewers' comments and remarks regarding the article: “Luminescence properties of KLaF4 doped with Pr3+ ions under VUV and visible excitation” and the authors' answers.
We would like to thank all reviewers for their insightful work. We believe that a good article is the result of the joint work of authors and reviewers. It is thanks to their comments that the article could be improved.
All reviewer comments were addressed. Changes we made to the manuscript are marked in yellow highlighted text.
Best Regards,
- Deren on behalf of all authors
Responses to Reviewer 3:
Reviewer's comments:
The article reports the introduction of praseodymium in KLaF4 crystals. The material was characterized concerning the photoluminescence properties and the application as thermometer.
The fact that is the first report of this material makes the work with potential to be published.
But I only can agree with the publication after addressing some questions.
These are the main questions:
- “The wide band beginning at around 71,000 cm-1can be attributed to host absorption.” The spectra only show what may be the beginning of the band. Did the authors performed emission spectra excited at a wavelength between 120 and 130 nm, to be sure that the wide band is not an artifact?
Response to Reviewer 3, comment No. 1:
Due to the lack of a proper excitation source, we did not perform emission spectra excited at a wavelength between 120 and 130 nm. We concluded that this band may correspond to host absorption due to its high energy (about 8.8 eV). However, we cannot exclude the fact that the origin of this band is different. Because of that, we changed this part of the manuscript as follows:
“The broad band, commencing around 71,000 cm-1, could be associated with host absorption, though its other potential origins cannot be excluded. Further detailed research is necessary to confirm the assignment of this band.”
- Line 231: “The highest integral intensity of visible luminescence was obtained for the sample doped with the 0.5% of activator ions” Emission spectra is not a quantitative technique, the intensity of different samples can not be compared.
Response to Reviewer 3, comment No. 2:
The emission spectra of samples with different Pr3+ ion concentrations were measured using the FLS1000 fluorescence spectrometer, maintaining the same set-up for excitation and emission slits, number of scans, integration time, step, and holder positions for all measured samples. To calculate the integral intensity, all recorded spectra were normalized and integrated into the range 455 – 800 nm. The obtained results are not absolute values but relative intensities. All calculated integral intensities were normalized to the highest value, which was obtained for the sample with a 0.5% concentration.
Although a single emission spectrum cannot be a quantitative technique, spectra recorded for samples of the same materials with different activator concentrations under the same measurement conditions can be used to make a relative comparison of luminescence intensity. This approach can be found in many papers, i.e.,:
- Stefańska, D.; Dereń, P. J. High Efficiency Emission of Eu2+ Located in Channel and Mg‐site of Mg2Al4Si5O18 Cordierite and Its Potential as a Bi‐Functional Phosphor Toward Optical Thermometer and White LED Application. Optical Mater. 2020, 8. 2001143.
- Kumar, P.; Singh, D.; Gupta, I. Gadolinium-based Sm3+ activated GdSr2AlO5 nanophosphor: synthesis, crystallographic and opto-electronic analysis for warm wLEDs. RSC Adv. 2023, 13, 7703–7718.
- Wang, Q.; Mu, Z.; Zhang, S.; Zhang, Q.; Zhu, D.; Feng, J.; Du, Q.; Wu, F. A novel near infrared long-persistent phosphor La2MgGeO6:Cr3+, Re3+ (Re = Dy, Sm). Lumin. 2019, 206, 618–623.
- Müller, M.; Fischer, S.; Jüstel, T. Luminescence and energy transfer of co-doped Sr5MgLa2(BO3)6:Ce3+, Mn2+. RSC Adv. 2015, 5, 67979–67987.
To emphasize the fact that the presented data corresponds with the relative intensities, we changed the y-axis label in the inset in Figure 5b and rephrased the text:
“To investigate the influence of dopant concentration, emission spectra were acquired under 444 nm excitation for samples with varying Pr3+ ion concentrations but under the same measurement conditions. The highest relative intensity of visible luminescence was obtained for the sample doped with the 0.5 % content (see inset in Figure 5b).”
- How the relative sensitivity values compare with other values reported for thermometers based on lanthanide ions?
Response to Reviewer 3, comment No. 3:
According to the literature, the condition for the high thermal relative sensitivity material is the SR > 1 %K-1 [1]. In our work, the highest SR was found to be 1.70 %K−1 at T = 140 K and 1.45 %K−1 at T = 175 K for the (3P1 → 3H4/3P0 → 3H4) and (3P1 → 3H5/3P0 → 3H5) transition ratios respectively. Due to the wide variety of reported luminescence thermometers based on Ln3+ ions, we decided to compere the materials that have the highest sensitivity in the low temperature range. In the bellowed table, we listed the SR MAX values reported for different hosts.
Host |
Dopant |
TMAX [K] |
SR MAX (%K-1) |
Reference |
KLaF4 |
Pr3+ |
140 |
1.70 |
This work |
β-NaYF4 |
Pr3+ |
120 |
≈ 5 |
[1] |
Y2O3 |
Nd3+ |
123 |
1.51 |
[2] |
NaYF4 |
Nd3+ |
203 |
16.3 |
[3] |
fluoroindate glass |
Er3+ |
152 |
2.8 |
[4] |
NaGdF4 |
Yb3+ |
125 |
≈ 1.2 |
[5] |
La2MgTiO6 |
Cr3+, V4+ |
165 |
1.96 |
[6] |
[GA]Mn(HCOO)3 |
Cr3+ |
100 |
1.20 |
[7] |
(Me2NH2)3[Eu3(FDC)4(NO3)4]·4H2O |
Eu3+ |
170 |
2.7 |
[8] |
The highest SR value of KLaF4:Pr3+ is comparable with the other materials working in the same temperature range. However, there are compounds, such as NaYF4:Nd3+ or fluoroindate glass doped with Er3+, that exhibit better thermometric performance.
On the other hand, we believe that due to the thermal coupling of the 3P0 and 3P1 levels, Pr3+-doped latticed can be used to design primary luminescence thermometers, which avoid the need for prior temperature calibration [10].
We added the above table to the Supporting Information as Table S4.
- Brites, C. D.; Balabhadra, S.; Carlos, L. D. Lanthanide‐based thermometers: At the cutting‐edge of luminescence thermometry. Opt. Mater. 2018, 35, 2302749.
- Zhou, S.; Jiang, G.; Wei, X.; Duan, C.; Chen, Y.; Yin, M. Pr3+-Doped β-NaYF4 for Temperature Sensing with Fluorescence Intensity Ratio Technique. Nanosci. Nanotechnol. 2014, 14, 3739–3742.
- Kolesnikov, I. E.; Kalinichev, A. A.; Kurochkin, M. A.; Mamonova, D. V.; Kolesnikov, E. Y.; Lähderanta, E.; Mikhailov, M. D. Bifunctional heater-thermometer Nd3+-doped nanoparticles with multiple temperature sensing parameters. Nanotechnology 2019, 30, 145501.
- Trejgis, K.; Ledwa, K.; Bednarkiewicz, A.; Marciniak, L. A single-band ratiometric luminescent thermometer based on tetrafluorides operating entirely in the Infrared Region. Nanoscale Adv. 2022, 4, 437–446.
- Haro-González, P.; León-Luis, S. F.; González-Pérez, S.; Martín, I. R. Analysis of Er3+ and Ho3+ codoped fluoroindate glasses as wide range temperature sensor. Res. Bull. 2011, 46, 1051–1054.
- Zheng, S.; Chen, W.; Tan, D.; Zhou, J.; Guo, Q.; Jiang, W.; Xu, C.; Liu, X.; Qiu, J. Lanthanide-doped NaGdF4 core-shell nanoparticles for non-contact self-referencing temperature sensors. Nanoscale 2014, 6, 5675–5679.
- Stefańska, D.; Bondzior, B.; Vu, T. H.; Grodzicki, M.; Dereń, P. J. Temperature sensitivity modulation through changing the vanadium concentration in a La2MgTiO6:V5+,Cr3+ double perovskite optical thermometer. Dalton Trans. 2021, 50, 9851–9857.
- Stefańska, D.; Kabański, A.; Vu, T. H.; Adaszyński, M.; Ptak, M. Structure, luminescence and temperature detection capability of [C(NH2)3]M(HCOO)3 (M = Mg2+, Mn2+, Zn2+) hybrid organic–inorganic formate perovskites containing Cr3+ Sensors 2023, 23, 6259.
- Li, L.; Zhu, Y.; Zhou, X.; Brites, C. D.; Ananias, D.; Lin, Z.; Paz, F. A.; Rocha, J.; Huang, W.; Carlos, L. D. Visible‐light excited luminescent thermometer based on single lanthanide organic frameworks. Funct. Mater. 2016, 26, 8677–8684.
- Maturi, F. E.; Gaddam, A.; Brites, C. D.; Souza, J. M.; Eckert, H.; Ribeiro, S. J.; Carlos, L. D.; Manzani, D. Extending the Palette of Luminescent Primary Thermometers: Yb3+/Pr3+ Co-Doped Fluoride Phosphate Glasses. Mater. 2023, 35, 7229–7238.
- Can the authors calculate the temperature uncertainty? This would be important to determine the operation range of the thermometer.
Response to Reviewer 3, comment No. 4:
Thank you for the valuable comment. We calculated the temperature uncertainty and added the following part to the manuscript:
“Temperature resolution δT (uncertainty in temperature), which refers to the minimal temperature change that causes significant fluctuation in an examined parameter, was estimated according to the formula [6]:
(15)
where δΔ/Δ represents the relative error in determining the thermometric parameter. (around 0.5 % for emission ratiometric measurements [7]). For the (3P1 → 3H4/3P0 → 3H4) and (3P1 → 3H5/3P0 → 3H5) ratiometric estimations, relative sensitivity with temperature resolutions of 0.29 K and 0.34 K over the whole measurement range can be estimated, respectively.”
- Lisiecki, R.; Macalik, B.; Komar, J.; Berkowski, M.; Ryba-Romanowski, W. Impact of Temperature on Optical Spectra and Up-Conversion Phenomena in (Lu0.3Gd0.7)2SiO5 Crystals Single Doped with Er3+ and Co-Doped with Er3+ and Yb3+. J. Lumin. 2023, 254, 119495.
- Wang, X.; Liu, Q.; Bu, Y.; Liu, C.-S.; Liu, T.; Yan, X. Optical Temperature Sensing of Rare-Earth Ion Doped Phosphors. RSC Adv. 2015, 5, 86219–86236.
- Did the authors measure the emission quantum yield?
Response to Reviewer 3, comment No. 5:
Although we are aware that quantum yield (QY) is an important parameter in the case of quantum-cutting phenomena, we did not perform QY measurement because of the technical difficulties. The most challenging problem in determining the QY of KLaF4:Pr3+ is the high excitation energy needed to excite the 5d levels and observe quantum cutting. The lowest Pr3+ 5d level has energy around 53,000 cm-1 (189 nm), which means that the excitation source must work in the vacuum conditions. In our, work we used the McPherson spectrometer equipped with a deuterium lamp which allows us to record the emission and excitation spectra in vacuum. However, there is no possibility to link this spectrometer with the integrating sphere and measured QY. Any other spectrometers that are equipped in the integrating sphere do not allow for sample excitation with energy higher than 200 nm. Moreover, we do not have a reference sample of known QY values to use the relative method. Although QY > 1 is a confirmation of the quantum cutting occurrence, we claim that the observed 1S0 → 1I6 and 3P0 → 3H4 transitions under 160 nm excitation are enough evidence.
Other minor questions:
In line 146 “photoluminescence spectrum” should be replace by “emission spectrum”, considering that an excitation spectrum is also a photoluminescence spectrum.
Response:
We accepted the comment and changed “photoluminescence” to “emission spectrum”.
“VUV” is used without being defined.
Response:
Thank you for pointing this out. The oversight was corrected.
Characters in some figures are too small.
Response:
We enlarged the font size in some Figures if it was possible.
In line: “The arrow indicates the excitation energy.” I understand that the wavelength was used for measured the spectrum in Figure 3b. But I do not see any advantage of having the arrow in the spectra, since that excitation wavelength is not relevant for the spectrum in Figure 3a.
Response:
Thank you for your comment. We agree that the arrow in the spectrum in Figure 3a is not necessary; however, we would like to keep it in the graph for two reasons. First, because the low x-axis is expressed as energy in cm-1, the arrow is helpful in localizing the energy of the excitation light. Second, although the excitation wavelength used for emission measurement is not relevant to the highest excitation band, we wanted to emphasize that 160 nm radiation allows for the excitation of the highest 5d levels of Pr3+ ions. Both of this information could be noticed without an arrow, but due to the importance of these aspects, we decided to stress them with an arrow. We completed the caption of Figure 3a:
“Figure 3. a) Excitation spectrum of the KLaF4:1%Pr3+ sample monitored at 272 nm. The arrow indicates the excitation energy used for emission measurement.”
Why the author chose 160 nm for the emission spectra if that is not the maximum of the excitation spectrum?
Response:
For emission spectra measurements, we utilized the deuterium lamp, which has the highest luminescence intensity around the 160 nm wavelength. The below spectrum presents the emission of the deuterium lamp used for recording the spectrum in Figure 3b.
The Intensity of D2 lamp emission is over ten times higher for 160 nm than for 184 nm (maximum of excitation KLaF4:Pr3+). As a result, we decided to use the 160 nm wavelength to obtain an intense and good-quality emission spectrum. At the same time, it should be emphasized that the excitation spectrum was corrected for the spectral characteristics of the D2 lamp.
“All of them are characterized by small values of FWHM (Full Width at Half Maximum) (see Table S2)”. They are small when compare with what?
Response:
The sentence was rephrased as follows:
“They all exhibit significantly smaller FWHM values, as indicated in Table S2, in comparison to the bands observed in the excitation spectrum. This implies that they align with 4f → 4f transitions.”
Reference is needed at the end of line 225.
Response:
The below reference was added:
[40] Runowski, M.; Woźny, P.; Martín, I.R.; Lavín, V.; Lis, S. Praseodymium Doped YF3:Pr3+ Nanoparticles as Optical Thermometer Based on Luminescence Intensity Ratio (LIR) – Studies in Visible and NIR Range. J. Lumin. 2019, 214, 116571.
That was the temperature used to measure the data in Figure 3 and 5?
Response:
Spectra and decay kinetics presented in Figures 3 and 5 were recorded at T= 300 K. We added the missing information to the figures and their captions.
There is an error in the line 17 of the supplementary material.
Response:
Thank you for pointing out the mistake. We changed “on excitation and emission spectra” to “in the excitation and emission spectra”.
Reference is needed at the end of line 398.
Response:
The below reference was added:
[50] Lisiecki, R.; Macalik, B.; Komar, J.; Berkowski, M.; Ryba-Romanowski, W. Impact of Temperature on Optical Spectra and Up-Conversion Phenomena in (Lu0.3Gd0.7)2SiO5 Crystals Single Doped with Er3+ and Co-Doped with Er3+ and Yb3+. J. Lumin. 2023, 254, 119495.

Round 2
Reviewer 3 Report
Comments and Suggestions for Authors
The authors answered all my questions and modified the manuscript according to my suggestions, so I agree with the publication.